# A Vaccine against Cancer: Can There Be a Possible Strategy to Face the Challenge? Possible Targets and Paradoxical Effects

**DOI:** 10.3390/vaccines11111701

**Published:** 2023-11-08

**Authors:** Roberto Zefferino, Massimo Conese

**Affiliations:** 1Department of Medical and Surgical Sciences, University of Foggia, 71122 Foggia, Italy; 2Department of Clinical and Experimental Medicine, University of Foggia, 71122 Foggia, Italy; massimo.conese@unifg.it

**Keywords:** metastasis, immune cells, cancer-associated fibroblasts, tumor-associated macrophages, TGF-β, epithelial–mesenchymal transition, tumor microenvironment, vaccines, immunotherapy, paradoxical results

## Abstract

Is it possible to have an available vaccine that eradicates cancer? Starting from this question, this article tries to verify the state of the art, proposing a different approach to the issue. The variety of cancers and different and often unknown causes of cancer impede, except in some cited cases, the creation of a classical vaccine directed at the causative agent. The efforts of the scientific community are oriented toward stimulating the immune systems of patients, thereby preventing immune evasion, and heightening chemotherapeutic agents effects against cancer. However, the results are not decisive, because without any warning signs, metastasis often occurs. The purpose of this paper is to elaborate on a vaccine that must be administered to a patient in order to prevent metastasis; metastasis is an event that leads to death, and thus, preventing it could transform cancer into a chronic disease. We underline the fact that the field has not been studied in depth, and that the complexity of metastatic processes should not be underestimated. Then, with the aim of identifying the target of a cancer vaccine, we draw attention to the presence of the paradoxical actions of different mechanisms, pathways, molecules, and immune and non-immune cells characteristic of the tumor microenvironment at the primary site and pre-metastatic niche in order to exclude possible vaccine candidates that have opposite effects/behaviors; after a meticulous evaluation, we propose possible targets to develop a metastasis-targeting vaccine. We conclude that a change in the current concept of a cancer vaccine is needed, and the efforts of the scientific community should be redirected toward a metastasis-targeting vaccine, with the increasing hope of eradicating cancer.

## 1. Introduction

Despite the subject of cancer vaccines was being approached using different strategies with the purpose of reducing the cancer mass and helping chemotherapy and radiotherapy, little attention has been paid to another important issue that has been underrated in the cancer field, i.e., the formation of metastases. Indeed, malignant tumors share some features, among which the most important is certainly the ability to metastasize, i.e., the capability that some cancer cells have to migrate to distant organs and tissues, where they reconstitute a neo-formation similar to the primitive tumor.

Nowadays, we can eradicate primitive tumors when they are still small, and surgical techniques have made a lot of progress; for example, in cases of breast cancer, the surgeon is capable of eliminating only a breast quadrant, saving the healthy tissue. Unfortunately, when surgeons intervene, they cannot be sure that the patient will be free of the disease forever.

Thus, metastatic lesions cannot be completely cured using current therapeutic approaches; in fact, about 90% of death in cancer patients is attributable to the appearance and development of metastases [1]. For this reason, the more we understand the metastatic process, the greater the chance of overcoming this problem. Presently, cancer vaccines are a form of immunotherapy that can help educate the immune system to recognize and eliminate tumor cells. The only preventive vaccines approved by the U.S. Food and Drug Administration (FDA) are those against HBV and HPV infections, and they function in exactly the same way that antiviral vaccines do. These vaccines help prevent the development of HPV-related anal, cervical, head and neck, penile, vulvar, and vaginal cancers, as well as HBV-related liver cancer. Therapeutic cancer vaccines are presently represented by only two formulations, which are also approved by the FDA: the sipuleucel-T vaccine, used for the treatment of patients with advanced prostate cancer, and Bacillus Calmette–Guérin (BCG), used for the treatment of early-stage bladder cancer [2]. The use of a few vaccines in cancer prevention/treatment underlies what is well known in the field, i.e., that there is not only one type of cancer; in fact, cancer constitutes many diseases, each characterized by its own distinguishing antigens. As a result, more sophisticated cancer vaccine approaches are necessary. In this review, we would like to draw attention to another important issue that, in the cancer vaccine field, has been underrated: metastasis.

We reviewed the scientific literature concerning metastasis, although not in a systematic way. We created a narrative review; so, we tried to tie the threads of the skein back together, not to unravel it.

## 2. Preliminary Considerations Regarding Carcinogenesis and Metastatic Processes

Usually, a vaccine stimulates the immune system toward an antigen, thus preventing the occurrence of a disease. This strategy is easily achievable when we know the cause of the disease being treated, as occurs in the case of infectious diseases. Such diseases are determined by a single agent, favoring the process of vaccine manufacturing: in fact, once we have identified the gene/protein capable of inducing an appropriate and defensive immune reaction in order to achieve our goal, the disease does not occur.

If we knew the protein or, generically, the substance that causes cancer, we would be near our endpoint. In this article, we would like to explain that, to date, nothing has been discovered in this respect. In fact, numerous different mechanisms/events can sometimes act as an enemy—and other times, an ally—to growth. From this perspective, individuating the correct and decisive therapy seems impossible, especially when we observe that these mechanisms use the same substances and pathways in opposite ways during the carcinogenesis process.

Thus, our narrative review will begin by addressing one point directly. We would like to highlight that the results of the scientific literature seem to be not only contradictory but also paradoxical. Thus, creating a vaccine capable of preventing metastasis, as we propose, appears difficult because, on the one hand, the metastatic process is a scarcely researched field; on the other hand, observing the scientific literature shows that there are plenty of different paradoxical results.

One explanation could derive from the complexity of the disease and from the high number of systems and mechanisms involved in homeostasis: cancer is capable of both knocking them out and using them to facilitate malignant progression. Another issue is that it could be problematic to reconcile the results coming from different experimental models.

As a simple example, in order to underline these paradoxical effects, we cite the behavior of gap junction intercellular communication (GJIC). It is known that in the early phases of cancer, the inhibition of GJIC favors proliferation (promoter effect); then, in the last phases, the inhibition of GJIC prevents metastasis. Its facilitation can determine the start of the metastatic process [3].

Starting from these considerations, we would like to verify and point out what has just been affirmed: analyzing the different players in the metastatic process, we will consider, in particular, the tumor microenvironment (TME), the extracellular matrix (ECM), TGF-β, the epithelial–mesenchymal transition (EMT), cancer-associated fibroblasts (CAFs), and the metastatic niche. Then, we will add two paragraphs regarding the important links between immunity/inflammation and cancer, as well as between acquired resistance to therapy and the TME; our main intention is to supply useful prospects to acquire new knowledge and provide nourishment for new thoughts and ideas.

### 2.1. The Tumor Microenvironment

Ursini-Siegel J. and Siegel P.M. [4] highlighted that the concept of the microenvironment is important not only in the first phases of cancer when numerous mediators act but also during the metastatic process. It is useful to add that the first phases of cancer are quite well known, although the results are not exhaustive; with regard to metastasis, Paget proposed the “seed and soil” theory in 1889 [5], which has been revitalized after more than 100 years [6].

Cancer cells move from the primary site to distant organs via blood and lymphatic vessels [7,8]. Before entering the circulation, cancer cells need to detach from tissue and invade it, whereas once tumoral cells have reached the target site, they have to extravasate and colonize it, giving rise to metastasis [9]. Importantly, during these different phases, various components of the tumor microenvironment (TME) at both the primary site and the metastatic site interact, including immune cells and stromal cells, as well as extracellular mediators, such as chemokines and cytokines. An important and complex crosstalk occurs between tumor cells and the above-mentioned cells that constitute the TME, controlling and affecting both tumor growth and metastasis [10,11,12]. Although targeted therapy and immunotherapy have also been implemented to target metastasis [13], the limited knowledge of the underlying mechanisms of cancer metastasis has resulted in the failure of our strategies to prevent and inhibit cancer metastasis in most cases.

To argue for our overall point, it will be useful to briefly consider the numerous mechanisms involved in metastasis, pointing out not every facet of this topic but only the paradoxical results from the scientific literature. There is no doubt that the first phase of metastasis consists of the ability of cancer cells to lose contact with the surrounding matrix, i.e., the basement membrane (BM) and ECM, and the following phases consist of the migration and invasion of other organs. To achieve this goal, cancer cells have to enter the blood or lymphatic circulation, where they will be subjected to different noxae, of which the main ones are anoikis and immunosurveillance [14]; then, just a few cancer cells will be able to overcome these hostile conditions. Once cancer cells enter the secondary organ, they adhere to the endothelium of the site in the so-called “pre-metastatic niche” [15,16,17]. Here, they can remain dormant as single cells or form numerous cancer cells, which give rise to micrometastases that will eventually become clinical metastases [18,19].

Different factors play a role in the metastatic process, but the most important ones involve genetic and epigenetic modifications at the level of tumor-initiating cells, also known as cancer stem cells (CSCs); immune escape; and TME changes [9,13].

The key players in the metastatic process are located in the TME: we should consider both cells (immune cells, CAFs, mesenchymal stem cells (MSCs), and endothelial cells) [20,21,22] and extracellular components, which mainly include chemokines, cytokines, tumor-secreted extracellular vesicles (EVs), and growth factors [23].

Importantly, not all of these players should be considered distinct compartments because they control and influence each other, constituting a mutual feedback network; in fact, they behave as a whole. Below, we will give a glimpse of this concept. Paradoxical notions concerning cells and mediators have arisen from the literature, and we have based this review on the available data (Figure 1 and Figure 2). The appraisal and development of vaccine targets stem from these paradoxes. In spite of the considerable complexity regarding the role of the TME, as shown in Figure 1 and Figure 2, we try to identify its key roles, and the figures allow us to easily illustrate the paradoxical effects that make it very difficult to create a cancer vaccine. Cancer cells have both intrinsic and extrinsic factors: it is well known that the cells also communicate with each other using gap junctions; in particular, similar and different cells establish connections using homologous or heterologous GJIC, respectively. During the metastatic process, when connexins (Cxs) act as tumor enhancers, it has been indicated that, in the case of GJIC between cancer cells and cells of the TME, Cxs may increase motility (such as glioma cells communicating with astrocytes); furthermore, they promote intravasation and extravasation processes through GJIC-independent mechanisms, mediated by their C-terminal (CT) domains [24]. When isolated in the extracellular matrix, Cxs can act on motility either independently of GJIC or as a result of the interaction of the Cx CT with the actin cytoskeleton. According to these findings, Cxs may favor metastasis through their involvement in the formation of invadopodia and the secretion of proteases during the invasion process. The authors propose verifying the connexin/pannexin ratio in the tumor, because it could allow the differentiation of the opposite effects of connexins. Through intercellular communication, we can probably verify the role that the TME plays during the metastatic process, allowing the cells to coordinate with each other. The complexity of this player is highlighted by the circumstance that all schematic mechanisms depicted in Figure 1 and Figure 2 are regulated by both intercellular communication and not-well-known ECM-remodeling mechanisms.

### 2.2. The ECM

Different signaling pathways regulate the metastatic process; however, the role of the ECM cannot be neglected, since it is involved in cancer cell migration [13]. In addition to this role, a relationship between the ECM and the tumor vasculature has been found in the context of tumor resistance to therapy [25]. The role of the pressure of the blood circulation through interstitial areas, affecting the transport of drugs to the tumor tissues, seems to be very important. The desmoplastic stroma has been found to be able to cause drug resistance, being able to affect the vascular systems in tumors and thereby impede the transport of anti-cancer drugs [26]. In particular, the organization of the ECM increases the fluid pressure due to the growth of the tumor mass, and this reduces the efficacy of drug delivery [27,28].

The ECM is composed of an acidic milieu due to glycolysis, since the tumor prefers the latter rather than oxidative metabolism; in fact, it converts glucose to lactate for the production of ATP, resulting in low-pH microenvironments [29,30,31]. Since tumors present a jeopardized vasculature, giving rise to hypoxia and the decreased removal of acidic products, there is a further tendency toward low-interstitial-pH products [30,32]. A possible hypothesis is that oncogenic pathways and genetic instability are caused by these hypoxic and acidic conditions, favoring tumor development and resistance [33].

An important factor inducing resistance to cancer cell therapies is the adhesion of cancer cells to the ECM [34]. Interactions between integrins and ECM components such as collagen, fibronectin, and laminin play an important role in cell-adhesion-mediated drug resistance [35,36,37,38,39].

For tumor therapy, impeding the communication between tumor cells and the TME has been proposed. To achieve this purpose, the targeting of adhesion molecules, proteolytic enzymes, and ECM components was conceived, a strategy that often represents an efficient option [40]. However, a few studies on glioblastoma, melanoma, and prostate tumors seem to suggest that several tumors with different extracellular matrices can show different therapeutic responses, indicating that targeting the ECM has limited outcomes in malignancies in late stages [41,42,43].

The role of the TME that occurs through ECM structures appears important; specifically, using intracellular pathways, it can drive the cell toward dormancy or proliferation. It has been shown in squamous cell carcinoma cells (HEp3) that the downregulation of the urokinase receptor inactivated α5β1-integrins, with the consequent inhibition of focal adhesion kinase (FAK) and the binding of cells to fibronectin [44]. The opposite mechanism capable of inducing dormant cancer cells’ activation in the tumor microenvironment operates by activating signaling cascades, such as collagen I-mediated integrin β1 signaling and that of Src and FAK for the phosphorylation of the myosin light-chain kinase via the ERK pathway [45]. These rearrangements in the ECM composition seem crucial in determining dormancy or inducing tumor metastasization.

Drug delivery also seems to be influenced by the ECM by orchestrating the activity of cytokines, such as TGF-β. This cytokine recruits fibroblasts to the tumor site and then transforms them into CAFs by inducing ECM matrix degradation [46,47,48]. Importantly, TGF-β and HIF-1 induce lysyl oxidase (LOX), an enzyme that causes cross-linked collagen and thus increases ECM stiffness [25,49,50]. Upagupta et al. [51] and Najafi et al. [52] demonstrated that ECM stiffness is a powerful inducer of TGF-β, with the result being the establishment of a bridge in the basement membrane, which favors tumor cell evasion. Hugo et al. showed through genomic and transcriptomic analyses that TGF-β activates gene sets, which, in addition to EMT, also regulate angiogenesis, wound healing, and dissemination [53].

Theoretically, LOX could be a target for a vaccine capable of preventing metastasis, even though it shows the opposite behavior depending on the tumor O_2_ concentration (see Figure 1).

### 2.3. TGF-β

It is well known that various pathophysiological processes, including the epithelial–mesenchymal transition (EMT), wound healing, angiogenesis, and dissemination, are regulated by TGF-β signaling [53]. TGF-β, through CAFs, lymphocytes, and endothelial cells, can also orchestrate and regulate immune responses [37,54,55]. Another mechanism exerted by TGF-β is represented by the inhibition of the proliferation and differentiation of antitumor T cells by stimulating the generation of regulatory T cells (Tregs) [56]. We cannot neglect to mention that TGF-β induces the secretion of monocyte chemoattractant protein-1 (MCP-1), upregulating the expression of mesenchymal markers and chemotactic factors (CCL-2, 7, 8, 13), which are associated with tumor progression and immunosuppression [57,58]. In line with our tour of cancer development/progression and the paradoxical results, Imamura and colleagues verified that TGF-β signaling has two distinct and opposite roles. In fact, during the early stage of carcinogenesis (cancer progression), TGF-β acts as a tumor suppressor; in the late stage (metastasis), TGF-β induces the EMT and invasion of cancer cells, accelerating metastasis [59]. In conclusion, TGF-β could be used as a vaccine target to prevent the metastatic process; however, it cannot be administered in the early phase of the disease due to its paradoxical effects (see Figure 2)

### 2.4. Epithelial-to-Mesenchymal Transition

Before analyzing this important process, we should specify that it will be evaluated above all considering its regulation systems, which involve numerous EMT-associated transcription factors (EMT-TFs): if there is no doubt that the respective genes determine their effect, apart from their epigenetic control, it should also be considered that EMT-TFs are controlled and counterbalanced by different signaling pathways, as will be cited below. Our aim is to verify the occurrence of paradoxical effects and thus select possible candidates for a vaccine. A key driver of the metastatic process is the epithelial–mesenchymal transition (EMT), which consists of a reversible transition process during which epithelial cells gain mesenchymal characteristics, reducing their epithelial properties [60]. The opposite mechanism, i.e., the mesenchymal-to-epithelial transition (MET), is represented by the reverse transformation of mesenchymal cells into an epithelial state at the level of the metastatic site [60,61].

It has been known since 1994 that TGF-β induces EMT in mouse mammary gland NMuMG cells, as shown by the reduction in epithelial markers, the increase in mesenchymal markers, and the reorganization of the cytoskeleton [62]. The elimination of the Tsk7L type I receptor modified this phenotype, showing its importance; in fact, without it, TGF-β would not be able to cause EMT [62].

In the EMT process, the reorganization of cell junctions and cytoskeleton represents a key characteristic. Among cell-to-cell junctions, gap junctions (GJs) play an important but controversial role in cancer. GJs are channels made of connexins, pannexins, and innexins, although connexins (Cxs) have been investigated more. Cxs show paradoxical behavior. For instance, it has been observed in colorectal cancer that they act as tumor suppressors, specifically Cx-43 and Cx-45 [63,64]. Whereas the survival of patients with prostate, breast, and colorectal cancers was increased through the overexpression of Cx-43 [63,65,66], it was reduced in patients with bladder cancer, esophageal squamous cell carcinoma, and oral squamous cell carcinoma [67,68,69]. In human osteosarcoma U2OS cells, GJs failed to inhibit EMT, while GJs between U2OS cells and normal human osteoblasts did, suggesting that GJs inhibit EMT through U2OS–osteoblast communication [70]. Furthermore, it was shown that GJ antagonists abrogated the progression of glioma normally induced by electrochemical communication through gap junctions [71]. Could GJs become the target for a vaccine in the future? To answer this question, the age of the specific tumor, i.e., the phase of its growth, should be considered because, as described above, GJs play different and opposite roles in the early and late phases of cancer.

EMT-associated transcription factors are key drivers of the EMT program, and, paradoxically, their effect can be oncogenic [72,73,74] but also tumor-suppressive [75]. Among EMT-TFs, Twist1 and ZEB1 were found to be overexpressed and downregulated in lung cancer, respectively [76], indicating that they would act in opposite ways (see Figure 1).

It is worth noting that the EMT program receives an important contribution from different EMT-TFs, which are able to form a complex regulatory network to coordinate it; thus, a single EMT-TF may rarely dictate cancer metastasis [77]. Thus, we can exclude the use of these factors as molecular targets of cancer vaccines. Moreover, EMT-TFs show posttranslational regulation; in fact, they can be activated or inactivated through phosphorylation or ubiquitination, and this circumstance would also exclude their use in a vaccine strategy. For example, because the phosphorylation of Snail via a GSK-3β-mediated mechanism in its first motif leads to its retention and activation, phosphorylation in its second motif is instead capable of favoring its ubiquitination and degradation [78]. On the other hand, if phosphorylation by the PAR-atypical protein kinase C (aPKC) occurs, the result will be the inhibition of the ubiquitination of Snail, thus promoting tumor metastasis [79]. Another example of EMT-TFs’ complex roles concerns ZEB. A switch from ZEB2 to ZEB1 may cause the progression of melanoma, indicating that these two ZEB proteins do not often function in the same manner under this condition and that ZEB1 acts in a paradoxical way [80].

The paradoxical effects that we can verify also include that of PRRX1, a more recently discovered regulator of EMT. In fact, some authors have shown that PRRX1 overexpression could favor a better prognosis in breast cancer patients [81], while other authors demonstrated that the upregulation of this EMT-TF was associated with a poor prognosis linked to increased EMT in colorectal cancer [82].

The role of TGF-β signaling is noteworthy because this factor displays both oncogenic and tumor-suppressive effects [83]. Indeed, it can stimulate both EMT and the proliferation of lung cancer [84]; on the other hand, if TGF-β is inhibited, this would induce a tumor-suppressive effect [85,86]. Paradoxically, it was shown that TGF-β acts as a cancer suppressor via a lethal EMT and promotes apoptosis through the remodeling of lineage-specific transcriptional networks [87].

In addition, for Hedgehog (Hh) signaling, paradoxical effects can be noted; indeed, whereas some authors verified a tumor-promoting effect [88,89], other authors showed the Hh-mediated inhibition of EMT and a tumor-suppressive effect [90].

YAP/TAZ are potent EMT inducers, promoting the progression of several cancers [91,92]. Hippo can phosphorylate YAP, leading to its inactivation [93,94]; for this reason, Hippo could be considered a cancer suppressor. Some authors showed that, in the liver, dephosphorylated YAP is translocated to the nucleus to induce the EMT and metastasis of hepatic cancer cells (HCCs) [95]. It is known that the excessive formation of filamentous actin induces the dephosphorylation of LATS, a downstream effector of Hippo, stabilizing YAP and thereby promoting liver metastasis [96]; because YAP can both increase and reduce the formation of filamentous actin, in theory, it could also be an oncosuppressor—another paradoxical effect.

Radical oxygen species (ROS) also show paradoxical effects, since they are known cancer inducers [31,97] and can activate both EMT [98] and MET (mesenchymal–epithelial transition) [99] but can also act as oncosuppressors, promoting ferroptosis and preventing tumor metastasis; in this case, it is noteworthy that the administration of antioxidants would promote cancer [100,101].

ROS can also act indirectly on EMT-TFs through the transcription factor Activator Protein-1 (AP-1). In the presence of oxidative stress, disulfide bonds between cysteines in different subunits of AP-1 are formed, thus reducing DNA affinity and altering its interaction with EMT-TFs [102,103,104]. Prolyl hydroxylase, which functions as a redox sensor, is also inactivated by ROS and promotes EMT and metastasis [105,106,107].

Another paradoxical effect was found for FOXO proteins. For example, FOXO4 promotes the production of SOD2, which protects via oxidation [108], while FOXO1 is able to promote EMT and metastasis by binding to the promoter of Twist [109]. Paradoxical effects are also reported regarding another FOXO family member, FOXO3a, which has been shown to abolish the β-catenin-mediated transcription of ZEB1, thus repressing EMT in prostate cancer [110]. Moreover, FOXO proteins also regulate EMT without classical EMT-TFs. Functioning as a competitive endogenous RNA (ceRNA), FOXO1 mRNA impedes the degradation of E-cadherin mRNA by miR-9, maintaining the expression of E-cadherin and then avoiding breast cancer metastasis [111].

A paradoxical behavior can be observed for glucose metabolism: usually, cancer cells use, above all, glycolysis; during EMT, cells instead prefer oxidative phosphorylation [112]. Alterations in lipid metabolism also compete to regulate the fate of cancer cells by favoring ferroptosis [113]; conversely, some lipid molecules, such as oleic acid, protect cancer cells from ferroptosis, promoting metastasis [100]. Opposite effects have been observed regarding the lipid metabolism enzymes ATP-cytrate lyase (promoting metastasis) [114,115] and acyl-Co A synthetase (sensitizing cancer cells to ferroptosis) [116].

Mounting evidence also shows the paradox of the important link between inflammation and the tumor through the EMT program. Huang et al. [77] affirm that “EMT program and inflammatory response are affected by each other thus forming complex network, which leads to cancer regression or progression” [117,118,119].

If the EMT program induces metastasis, it is conversely also true that the cancer cell, in its complete mesenchymal state, is very prone to ferroptosis. Unfortunately, a certain number of cancer cells are found in an intermediate state (intermediate EMT, also reported as hybrid EMT, partial EMT, or incomplete EMT), exhibiting good plasticity and heterogeneity, which certainly favor cancer progression and drug resistance. This detailed examination of EMT-TFs and their paradoxical effects represent a reason for the exclusion of some possible candidates for future vaccines and a reason to propose others, as will be discussed below.

### 2.5. CAFs

CAFs are the most frequent cell type harbored in the TME. The role of CAFs in the metastatic process appears to be complex, since they can educate the TME and cancer cells, favoring cancer proliferation and causing the presence of invasive and metastatic phenotypes [120,121,122]. Even with CAFs, we cannot neglect to mention paradoxical effects considering that CAFs can transfer high-energy metabolites (L-lactate, ketones, glutamine, and free fatty acids) to epithelial cancer cells via caveolin-1 (CAV-1) to promote cancer progression [123] in a host–parasite association. However, it has been observed that EMT may be induced by the loss of CAV-1 in the tumor stroma, which leads to activated TGF-β signaling [124,125,126].

Another paradoxical observation is the CAF-mediated EGFR signaling pathway, which seems to be essential for several cellular functions, including the maintenance of cancer stemness, cell proliferation and invasion, and metastasis [122]. In line with this, the paradoxical effects are remarkable, since a positive correlation was observed between the overall survival period of patients with several cancers and EGFR overexpression in tumor cells [127]. Interestingly, other authors have reported that EGFR overexpression in CAFs has no significant relation to the prognosis of patients with colorectal cancer [128]. Overall, these studies indicate that EGFR in CAFs is likely not the only key player, so it cannot be considered an independent prognostic factor for survival evaluation in patients with cancer.

Considering that the crosstalk between CAFs and cancer cells is important for the subsequent growth of tumors, it could be useful to design a new therapeutic agent that can affect this mechanism. In the past, many attempts were made to verify this effect by targeting either Hippo signaling, including its downstream effector YAP1, which regulates this crosstalk [129,130], or NF-κB signaling, which has been implicated in the immunosuppressive function of CAFs and in protumoral inflammation [122,131].

Considering the crosstalk between CAFs and immune cells (T cells, TAMs, and myeloid-derived suppressor cells (MDSCs)), it is known that these interactions are involved in the increase in tumor immunosuppression and could be the target of cancer therapies, in combination with other options. It can be mentioned that pirfenidone (PFD), a TGF-β blocker, has inhibitory effects on CAF migration and decreases the expression of PD-L1 by targeting CCL17 and TNF-β. In breast cancer, following PFD treatment, CAFs show a reduction in the occurrence of their immunosuppressive capacity, thus increasing the efficacy of chemotherapy [132]. Losartan (an angiotensin receptor blocker) was capable of driving myofibroblast CAFs into a quiescent state, thus attenuating immunosuppression and increasing T-lymphocyte activity; the result was an improvement in the response of breast cancer cells to immune checkpoint blockers [133]. These studies indicate that combination therapy driven by interventions targeting CAFs and immunotherapies might be an effective and promising strategy for treating resistant tumors.

Thus, the role of CAFs cannot be neglected here, and future therapeutic attempts should be made to transform these cells from enemies to allies of patients. Their role is manifold and is stimulated by agents in chemotherapy, whereby CAFs serve different functions in activating pro-survival signaling pathways, stimulating the proliferation, stemness properties, and autophagy of cancer cells, thus enhancing treatment resistance. Importantly, CAFs are able to change tumor cell metabolism, favoring cancer cell survival and preventing apoptosis due to treatment and/or stress. It was also shown that in lymphoma cells, CAFs promoted the survival of tumor cells by increasing the tricarboxylic acid cycle [134]. Moreover, CAFs could decrease cellular adhesion through the EMT program, which promoted both tumor metastasis and resistance [135].

Different markers allow us to distinguish activated CAFs from normal fibroblasts (NFs) [122]. Among these, fibroblast-activated protein (FAP) seems to be very important, so it has attracted the attention of the scientific community. Consequently, antibodies against FAP and other FAP-targeting drugs have been designed [136]. A couple of studies have also reported the fabrication of a DNA vaccine targeting FAP in order to deplete T-cell-mediated CAFs [137,138]. Although CAR-T cells directed toward FAP can kill CAFs, their use is associated with extended lethal osteotoxicity and cachexia linked to FAP expression typical of MSCs [139,140,141]. Attempts were made using nanoparticles based on ZnF16pc loading and FAP-specific single-chain variable fragments (αFAP-Z@FRT), which could target FAP-depleting CAFs both effectively and safely [142].

An interesting approach was suggested by Wu et al. [122], who, based on the observation that differentially expressed proteins in cancer cells and CAFs are common in tumor progression, proposed an alternative approach named sequential target perturbation (STP), which consists of first targeting CAFs to block the protumor effect and then treating cancer cells to bring about anti-cancer effects. This can be carried out sequentially: aiming at targeting, firstly, CAFs (such as targeting ECM remodeling to remove the barrier for CD8+ T-cell infiltration) and, secondly, cancer cells (such as performing PD-1/PD-L1 immune checkpoint therapy). In summary, these authors hypothesize that a promising CAF-driven immunotherapy should be based on many pillars: “ECM normalization, prevention of disturbances promoted by non-therapeutic immune cells, and combining other antitumor therapies with immunotherapy to maximize the efficacy of cancer immunotherapy” [122] (p. 24), thereby reducing the enemies and increasing the allies. To elaborate, a vaccine directed toward CAFs would be theoretically justified, but it cannot be recommended due to the important adverse effects mentioned above [136,137,138].

## 3. The Pre-Metastatic Niche

In this section, we will investigate the paradoxical behavior of cells, particularly immune cells, to verify whether they can be used as vaccine candidates. Many authors have proposed that the concept of a metastatic niche can explain the metastatic process, considering pre-existing and induced niches [4], where the role of immune cells seems to be essential: the main modulators of the pre-metastatic niche are infiltrating immune cells, insofar as they regulate diverse processes such as ECM remodeling, cancer cell adhesion, angiogenesis, and immune suppression.

Here, we encounter one example of the paradoxical behavior of cells and the function of metabolic pathways. In the case of neutrophils’ roles, in early phases, they are protective against the tumor (likely by inducing oxidative stress that limits cancer cell seeding) [143,144]; then, in other cases, they can permit, through the formation of extracellular NETs, the trapping of circulating cancer cells, helping to form the niche [145].

The complexity of the metastatic process is supported by experiments that show that neutrophils are toxic to cancer cells in the absence of TGF-β signaling, whereas, when the TGF-β pathway is functional, neutrophils are skewed toward an immunosuppressive phenotype, and the tumor grows [146].

Furthermore, the metabolic behavior of cancer cells can influence the formation of metastases. Indeed, numerous papers have verified that cancer can use more useful metabolic pathways to survive in novel environments; an example is the ability of mammary cancer cells to use low levels of glucose while extracting energy from glutamine when they metastasize to the brain [147].

An evident area where research could work to realize a vaccine against cancer is organotropism: studying and knowing in depth the mechanisms that regulate organotropism, which remains one of the most intriguing unanswered questions in cancer research, could lead to new candidate targets for vaccines.

The organotrophic metastasis pattern is probably due to the successful establishment of the pre-metastatic niche in specific organs, a process that transforms the local milieu of the target organ into a microenvironment favorable for the outgrowth of primary cancer cells. In this regard, an important role would be played by ECM remodeling. For example, in the lung, LOX activity is favored by a high oxygen concentration [148], while in other organs, LOX activity is induced by hypoxia through the activation of hypoxia-inducible factor (HIF), another paradoxical effect. The therapeutic use of LOX inhibitors has also generated paradoxical effects [149,150,151].

It is worth adding that the pre-metastatic niche has to be considered as a crossroads: here, cancer cells could become dormant or acquire a proliferative and aggressive capacity; thus, orienting cancer cells in this niche could be a therapeutic choice.

Therapeutic attempts have also been implemented with the anti-CD20 antibody (rituximab) or the anti-IL-2Rα antibody (daclizumab); they can bypass the tumor-evoked regulatory B-cell (tBreg)-mediated blockade of the immune response to some cancers [152]. However, it has been preclinically shown that rituximab does not provide benefits in solid tumors, as it can increase CD20^Low^ tBregs, thus depleting the beneficial antitumor CD20+ B cells, resulting in a further enhancement of lung metastasis by exacerbating tBreg-mediated immunosuppression [153]. This section may seem limited compared to the complexity of our argument, but our aim is to reveal possible candidates for vaccines that, by examining the literature, could be considered unworthy of attention. However, we considered it important to verify previously published works to provide new ideas about possible vaccine candidates.

## 4. Immunity and Inflammation

Observing the role of immune cells during carcinogenesis and linking their role in the course of acute and chronic inflammation, we explore these processes to reveal possible paradoxical behaviors that would exclude certain factors when designing specified vaccines (Figure 3). In this regard, it is worth citing the fact that various types of immune cells, as well as cytokines and chemokines, play a role in regulating the TME, inhibiting or favoring the metastatic process. The different roles of immune cells can be verified during carcinogenesis. In particular, in the first phase, innate immune cells phagocytose and destroy cancer cells due to their activation by lymphocytes, which differentiate into Th1 cells and secrete IFNγ, TNF-α, and IL-2. In later phases, chronic inflammation supports T cells’ differentiation into Th2 cells, which secrete cytokines that induce T-cell anergy, which can promote cancer development [154]. The evident role of Tregs also appears paradoxical. On the one hand, these cells can elicit neutrophils and macrophages to act, via cytokines, as a cancer-development dynamo; on the other hand, they facilitate dendritic cells and cytotoxic T cells with antitumor and metastatic effects [155,156]. Tregs regulate autoimmunity but also play an evident role in cancer progression, showing an inverse correlation with patient survival [157]. They can act through the suppression of an IFN-based antitumor response due to the production of Th2-polarizing cytokines (IL-10 and TGF-β) [158,159]. However, although anti-PD-1 therapy could play a role in preventing the tumor’s escape from the immune response, likely regulating the infiltration of Tregs, these studies did not reach statistical significance [160,161,162,163].

The useful evaluation of the TME could also concern the role of the polarization of tumor-associated macrophages (TAMs). It is well known that a pro-inflammatory state will promote an M1 response, which is highly efficient against cancer cells; conversely, an M2 response will favor immunosuppression. Concerning this possible transformation of M1 into M2 and vice versa, the TME plays an important role: within normoxic tumoral tissue, an M1 response is ensured, while in hypoxic regions, an M2 response would prevail [164,165].

Another example is related to interferons. The lack of type I IFN signaling in mice leads to an enhanced metastatic load and the accumulation of neutrophils in breast carcinoma lung and liver metastases [166], suggesting that it displays anti-metastatic actions. However, in the later stages of carcinogenesis, type I IFNs orchestrate an anti-immune program by inducing the expression of immunosuppressive factors (PD-L1, indoleamine 2, 3-dioxygenase (IDO), and IL-10) in dendritic cells [167,168] but can also facilitate EMT and promote inflammation at distal sites that enhance metastasis [169].

The chemokine network could also represent a field that we cannot neglect, since paradoxical results have also been observed for them [170]. For example, CXCL2, CXCL8, CXCL4, CXCL9, CXCL10, CXCL12, and CCL2 are proangiogenic, while CXCL9 and CXCL10, when combined with CXCL4, can inhibit angiogenesis [171]. Among them, the prevailing proangiogenic activity is pursued by CXCR2 [172], while CXCR3 is responsible for angiostatic actions [171]. Unfortunately, in this case, after verifying the literature, we again observe that the paradoxical behaviors of the cells and the effects of chemokines and cytokines, the pivot of inflammation and immunity, could exclude their use as candidates for vaccines that can prevent metastasis.

## 5. Acquired Resistance to Therapy and TME

After briefly giving an overview of the TME at the primary site and pre-metastatic niche, it would make sense to present data about the influence of these paradoxical issues on the cancer’s response to therapies. In accordance with the aim of developing a vaccine targeting metastasis, it could be useful to verify whether there is a possible candidate to develop a vaccine.

The mainstays of cancer treatment are chemotherapy, radiotherapy, and immunotherapy. However, most patients display poor outcomes due to resistance to them. Wu et al. [25], focusing on the TME as a target, highlighted that there are three mechanisms that are vehicles for acquired immune resistance: (i) increased levels of immunosuppressive cells and molecules; (ii) the upregulation of immune checkpoints; and (iii) tumor mutation loads and the loss of target antigens. A known example is that the recruitment of TAMs and MSDCs can reduce the sensitivity to immunotherapy and enhance immunosuppression, resulting in acquired resistance [173,174,175]. In this regard, Maj et al. [176] showed that Tregs undergoing apoptosis upregulate the extracellular adenosine concentration, associated, in mouse models, with acquired resistance to anti-PD-L1 mAb treatment. The compensatory inhibitory mechanism also evokes acquired resistance. As an example, in lung tumor biopsies, in both humans and mice, an association between increased CD4+ TIM-3+ and CD8+ TIM-3+ T cells and increased resistance to anti-PD-1 mAb was verified [177]. Reducing the extracellular adenosine concentration is a method to prevent acquired resistance but also to counteract immunosuppressive effects induced through the stimulation of A2a and A2b receptors, which are present in different immune cells. Thus, numerous attempts were made using an A2aR antagonist [178], specifically acting on ectonucleotidases (CD39 and CD73) that form adenosine via AMP. CD39 and CD73 could be targets for a vaccine to prevent acquired resistance; moreover, as adenosine induces immunosuppression, these ectonucleotidases could also be targeted for this purpose. In fact, high CD73 levels are associated with unfavorable clinical outcomes; however, there are reports that CD73 expression could mitigate the metastatic process [179,180,181,182]. In this case, we should also verify a paradoxical event, which is highly unexpected.

Although the roles of several cytokines always appear paradoxical (as discussed above), it has been envisioned that cytokines may play a role in cancer treatment when used along with other drugs, including immune checkpoint inhibitors and oncolytic viruses, or as a component of DC- and tumor-cell-based vaccines. Nevertheless, their clinical use is limited by a short half-life and systemic toxicity (pro-inflammatory and autoimmune reactions) [183].

In general, it is known that CAFs utilize adaptive mechanisms with tumor cells for acquired resistance in many cancers, including lung, breast, and prostate cancers and glioblastoma. As described above and sketched in Figure 1, various TME factors, including immune cells, tumor cells, and the ECM, interact with CAFs to participate in tumor cell resistance [25].

Another resistance mechanism is exerted by the dormancy of cells that have migrated into other organs. These cells, which are neither hyper-proliferating nor metabolically active, appear resistant to anti-proliferative therapy [184]. In accordance with recent research (reviewed in [185]), the dormant state depends on complex mechanisms that regulate the crosstalk between stem cells, cancer cells, and CAFs; nevertheless, a “genetic signature” of dormancy is lacking but worth finding, since it is linked to therapeutic resistance and tumor recurrence. Moreover, the complex interplay between cancer stem cells and immune cells [146,186,187,188,189,190,191,192,193,194,195,196,197,198] allows us to define these cell types as having a double capacity: they are players in resistance to common treatments, including immunotherapy, but they could be future targets in solid tumors [197,198,199]. Finally, a “recessive” mode of cell communication has been proposed with relevance to metastasis formation, which would be able to determine a dormant state in cells in a relatively suitable metastatic microenvironment [200]. If it were possible to counteract this condition, we could impair the metastatic process, since it permits cancer cells to resist chemotherapy.

## 6. Prophylactic and Therapeutic Cancer Vaccines

Besides those directed toward HPV and EBV, other prophylactic vaccines for cancers have not been found. Therapeutic cancer vaccines have also lagged behind. For example, sipuleucel-T, consisting of autologous peripheral-blood mononuclear cells that have been activated ex vivo with a recombinant fusion protein made of a prostate antigen called prostatic acid phosphate, fused to granulocyte-macrophage colony-stimulating factor, prolonged survival in men with metastatic castration-resistant prostate cancer, without a significant effect on the time to disease progression [201]. Notably, large phase III vaccine trials in patients with advanced cancer reported negative outcomes [202,203,204,205,206]. It is now becoming clear that the TME provides a unique environment that gives rise to the tumor properties of immune suppression and escape and therapy resistance (see above). Further studies have thus led to the understanding that cancer vaccines can be effective in the following conditions: (1) a low tumoral mass burden; (2) lowered or reduced immunosuppression via chemotherapy or immune checkpoint inhibitors; (3) a strong and sustainable specific immune response guided by CD4+ TH1 cells and CD8+ CTLs [207]. This also implies that neoantigens should be captured efficiently by APCs such as DCs and processed and presented on MHC-I and MHC-II molecules. All of the studies aiming to identify and use tumor antigens as therapeutic cancer vaccines should be informed by these notions.

A list of neoantigens that could be used for vaccines is given in recent literature reviews [207,208]. Our aim is to highlight that the molecules that are presented above in the TME theater and that gave paradoxical results have not been listed. The cause of this could depend on the fact that we focused our attention on the metastatic process and how it would be possible to counteract it; in contrast, previous attempts have focused on inducing the regression of the tumor mass as chemotherapy and immunotherapy try to do. We propose a different approach, i.e., the prevention of metastasis.

Nowadays, the strategies for improving cancer vaccine outcomes are as follows: (1) personalized neoantigen identification through whole-exome-sequencing, in silico, and machine-learning approaches [207,208]; (2) vaccines based on shared antigens in the case of a low tumoral mutational burden [207]; (3) in situ vaccines (ISVs), i.e., generating the vaccine itself in the TME and making therapeutic use of immunogenic cell death (ICD) [209,210], which, naturally or induced, makes tumoral cells release antigens and danger-associated molecular patterns, resulting in both the attraction/activation of immune cells and antigen capture by APCs and its presentation to T cells. Besides the BCG-based vaccine, which is already used in non-muscle-invasive bladder cancer as a TLR agonist [211], novel ISVs are also in clinical trials, but none of them are based on the paradoxical scheme presented above [207].

However, in our opinion, targets of cancer vaccines should be explored in terms of tumoral and TME mediators and cells that give paradoxical results and also, coincidentally, are responsible for intrinsic and extrinsic resistance mechanisms [212]. A tentative but not exhaustive list is given in Table 1.

## 7. Discussion

Following what we have tried to indicate above regarding the complexity of cancer, we can affirm that our current approaches to this disease seem mistaken. We have preferred a vaccine that aims to reduce the primary tumor mass, thus helping chemotherapy through the involvement of the immune system; however, this method could serve to win the battle but not the war. In fact, beyond the tumor-therapy-resistance mechanisms cited by various authors, our therapeutic attempts appear useless if metastasis occurs, since it can often result in a patient’s death.

If we change our approach and retain the right to develop a vaccine that prevents metastasis, we should find a path toward obtaining this result, even though we cannot ignore the fact that this new approach has to coincide with our limited knowledge of the phenomena involved in metastasization. The mechanisms involved often control each other and, as we observed above, often appear paradoxical. This reduces our chance of eradicating this disease even more. A useful approach would be to facilitate Th1 polarization to increase cellular responses to cancer cells; however, here, we must remark that type 1 IFN also has paradoxical effects, although its protumoral effect occurs only in later stages of carcinogenesis [167,168,169]. A consideration regarding our subject, which is marginal but important with respect to the causes of cancer, is the following question: should we prevent Th2 polarization induced, for example, by stress [254]?

Another area where we could intervene is the link between inflammation and cancer: there are many mechanisms involved that are mutually intertwined, but the crucial node seems to be the role of macrophages and CAFs. These cells are different from each other in many respects but share some features: they can be protumoral or suppressors, and all seem to depend on the features of the TME. For example, the reduction in the TME’s oxygen concentration can regulate the transformation of TAMs from M1 to M2 (see Figure 1), thus impeding the reduction in intratumoral oxygen concentration during growth, which would facilitate tumor suppression by M1 macrophages, impeding their transformation into M2.

Crosstalk between CAFs and cancer cells can also induce metastasis using TGF-β; in the future, will we be able to thoroughly understand this crosstalk? Will we be able to inhibit it? Which counterregulatory mechanisms would intervene?

Unfortunately, this review does not allow us to define specific strategies or recommendations for developing a metastasis-targeting vaccine: nowadays, this result seems too far to reach due to our limited knowledge regarding the metastatic process, along with the presence of numerous paradoxical behaviors or effects that hinder the development of an appropriate vaccine that can prevent metastasis. However, this article, which shows the complexity of the metastatic process and verifies different results in the literature that contradict each other, obliges us not to give up on deepening our knowledge of this important process while also using easily comparable experimental models that allow us to reach our goal in the future: to defeat metastasis and then cancer.

However, after reporting the paradoxical effects/behaviors of different players in metastasis and verifying the complexity of this process, we can consider the following:(a)As far as we know, it is impossible to create a vaccine that can protect healthy subjects, because the cause of cancer is not known, and there is probably no single cause toward which we could direct an immune reaction as a vaccine would be able to do.(b)Instead, it is possible to create a vaccine that is suitable for cancer patients because it is also true that we cannot prevent cancer, but we can prevent metastasis specifically in those patients who do not yet have it.(c)It is unquestionable that the first step is to locate the target, identifying the player of the metastasis and blocking it; the question that arises is whether there is a single player or many players.(d)Moreover, when should we intervene? Assuming that we were able to answer these questions, how could we know which condition the patient is in? In other words, what is the stage of the disease at the molecular level, considering that the paradoxical effects appear linked to specific periods in which one effect can be transformed into an opposite one?(e)With respect to the previous considerations, we propose some possible candidates for a vaccine against cancer:(1)An ectonucleotidase such as CD39: blocks the formation of adenosine;(2)LOX: an enzyme involved in ECM remodeling and promoting cancer cell invasion;(3)FOXO1: enables EMT in breast cancer;(4)PRRX1: its increase is associated with a poor prognosis in colorectal cancer;(5)Hh: it was shown to cause tumor progression in breast and cervical cancer;(6)TGF-β: only in the late stage.

Along with the possible candidates cited above, we could also discover different strategies that could prevent metastasis:−Establishing appropriate methods to increase O_2_ concentration in the TME, a feature that increases antitumoral M1 TAMs;−Finding substances that inhibit the crosstalk between CAFs and cancer cells;−Identifying specific strategies to increase IFN and generally stimulate the Th1 response.

In conclusion, the road is perhaps too long to answer these questions soon; nevertheless, it is important to start now by changing the current approaches to cancer vaccine strategies, with the aim of preventing the metastatic process.

## Figures and Tables

**Figure 1 vaccines-11-01701-f001:**
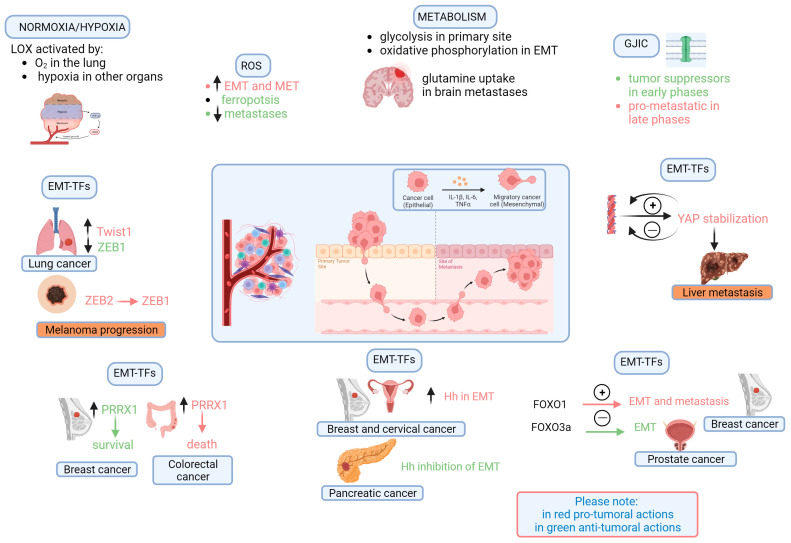
Paradoxical concepts about intrinsic cells and mediators within the TME. Normoxia/hypoxia. LOX, an enzyme that determines ECM remodeling and promotes cancer cell invasion, is promoted in the lung by high oxygen concentrations, while in other organs, it is induced by hypoxia through HIF-1 ROS. Oxidative stress is a well-known cancer inducer and can activate both EMT and MET; however, it can act as an oncosuppressor, promoting ferroptosis and preventing tumor metastasis. Metabolism. While the primary site is usually glycolytic (via the Warburg effect), during EMT, oxidative phosphorylation prevails; glutamine uptake, instead of glucose, has been revealed in brain metastases from breast cancer. GJIC. GJs mediating cell–cell communication are tumor suppressors in the early phases of carcinogenesis, but they are pro-metastatic in later phases, participating in the formation of the pre-metastatic niche. EMT-FTs. Twist1 and ZEB1 were found to be overexpressed and downregulated, respectively, in lung cancer. A switch from ZEB2 to ZEB1 promotes the progression of melanoma, suggesting that these two ZEB proteins function in an opposite manner and that ZEB1 acts paradoxically. PRRX1 overexpression is associated with a favorable prognosis in breast cancer patients, but its upregulation is related to a poor prognosis linked to increased EMT in colorectal cancer. The excessive formation of filamentous actin (F-actin) determines YAP stabilization, favoring liver metastasis. On the other hand, YAP can both increase and reduce the formation of F-actin, thus making YAP a likely oncosuppressor as well. Hh was verified to have a tumor-promoting effect on breast and cervical cancers and to inhibit EMT; however, a tumor-suppressive effect was shown in pancreatic cancer. FOXO factors have distinct effects on cancer: while FOXO1 can promote EMT and metastasis in breast cancer, FOXO3a represses EMT in prostate cancer. Other details and references can be found in the text. EMT: epithelial–mesenchymal transition; EMT-TFs: EMT transcription factors; GJIC: gap junction intercellular communication; Hh: hedgehog; LOX: lysyl oxidase; MET: mesenchymal–epithelial transition; ROS: reactive oxygen species.

**Figure 2 vaccines-11-01701-f002:**
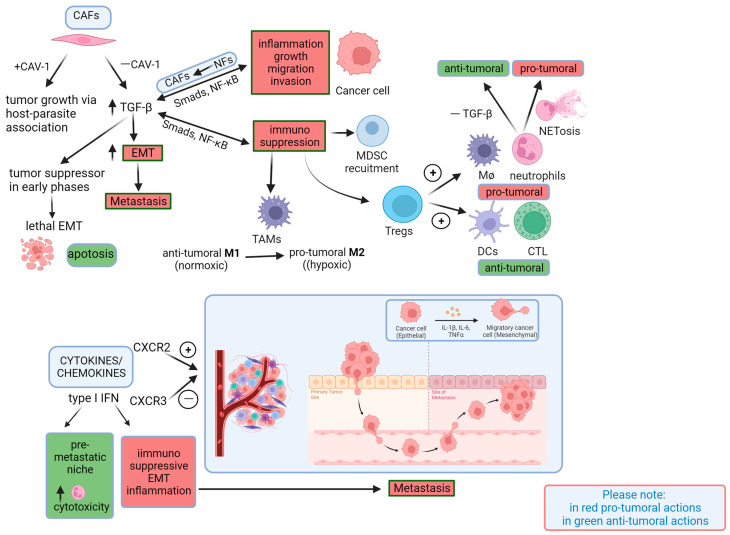
Paradoxical concepts about extrinsic cells and mediators within the TME. Cytokines/chemokines. Type I IFNs can inhibit the pre-metastatic niche by increasing neutrophil cytotoxicity in cancer cells and, in later stages of carcinogenesis, act as immunosuppressors, as well as EMT and inflammation inducers. Chemokines have both proangiogenic (mainly CXCR2) and anti-angiogenic (mainly CXCR3) effects. CAFs. In the presence of CAV-1, CAFs result in tumor growth via host–parasite association; in its absence, they secrete high levels of TGF-β, inducing EMT and metastasization. However, TGF-β can act as an oncosuppressor in early phases by inducing lethal EMT and apoptosis. Crosstalk between CAFs and cancer cells is mediated by CAF-derived TGF-β, which induces some cancer hallmarks (inflammation, growth, migration, and invasion), and conversely, tumor cells can induce NF transformation into CAFs through Smad and NF-κB signaling. Crosstalk between CAFs and immune cells also occurs through Smad and NF-κB signals, leading to immunosuppression. TAMs are switched from antitumoral M1 (in normoxia) to protumoral M2 (in hypoxia), and MDSCs are recruited. Tregs can have opposite effects: they can have protumoral effects by means of neutrophils and macrophages (Mø) and antitumoral effects by activating antigen-presenting cells (APCs), such as dendritic cells (DCs) and cytotoxic T cells (CTLs). Neutrophils can also act in a bimodal way: they are antitumoral in the absence of TGF-β and protumoral in its presence via undergoing NETosis (which entraps cancer cells in tumoral vessels). More details and references can be found in the text. CAFs: cancer-associated fibroblasts; CAV-1: caveolin-1; EMT: epithelial–mesenchymal transition; IFN: interferon; NET: neutrophil extracellular traps; NFs: normal fibroblasts; TAMs: tumor-associated macrophages.

**Figure 3 vaccines-11-01701-f003:**
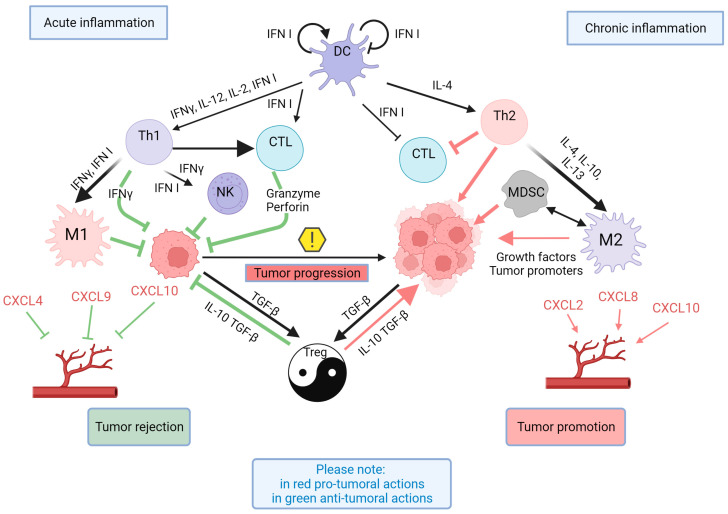
A schematic view of the contrasting roles of inflammation/immune responses during cancer progression. At tumor onset, a pro-inflammatory reaction prevails (left side), with DCs inducing Th1 cell polarization, with either direct or indirect antitumoral effects through the activation of CTLs and NK cells and the polarization of TAMs to the M1 type. During tumor progression, however, chronic inflammation occurs (right side), whereby tumor growth is promoted by Th2 CD4+ T cells and MDCS, which, in combination, both repress CD8+ cytotoxicity and induce the protumoral polarization of the innate immune response (such as M2 polarization of TAMs) via cytokine secretion (IL-4, IL-10, and IL-13). Treg cells have a dual role (yin and yang) in that, depending on the context, they have either antitumoral or protumoral effects via the secretion of IL-10 and TGFβ. On the other hand, cancer cells produce TGF-β, which can have protumoral or antitumoral Treg-mediated effects. Type I IFNs can activate antitumoral effects by autoregulating DCs and increasing the functions of T cells; however, chronic exposure to these cytokines has opposite effects. Different chemokines also have opposite effects on tumoral angiogenesis. CTL: cytotoxic lymphocyte; DCs: dendritic cells; IFN I: type 1 interferon; MDSC: myeloid-derived suppressor cell; NK: natural killer; TAM: tumor-associated macrophage: Treg: regulatory T cell.

**Table 1 vaccines-11-01701-t001:** Targets of cancer vaccines based on paradoxical roles and resistance mechanisms.

Intrinsic Paradoxes and Resistance		
Target	Paradoxes	
GJIC	Increased Cx expression: - Increased survival in prostate, breast, and colorectal cancers - Decreased survival in bladder cancer, esophageal squamous cell carcinoma, and oral squamous cell carcinoma	[63,65,66,67,68,69]
EMT-FTs	- Pro-metastatic - Tumor suppressor	[72,73,74,75]
TGF-β	- Stimulation of EMT and cancer cell proliferation - Cancer suppressor via a lethal EMT and promotion of apoptosis	[84,87]
Hedgehog pathway	- Induction of tumor progression - Inhibition of EMT and a tumor-suppressive effect	[88,89,90]
Hippo pathway	- YAP/TAZ can be both pro-metastatic and anti-metastatic through the regulation of F-actin	[91,92,96]
ROS	- As cancer inducers, they activate both EMT and MET - As oncosuppressors, they promote ferroptosis and prevent metastasis	[31,97,98,99]
FOXO proteins	- FOXO1 is able to promote EMT and metastasis - FOXO3a represses EMT	[109,110]
Cancer cell metabolism	- Glycolysis in initial stages - Oxidative phosphorylation during EMT - Use of glutamine instead of glucose in metastases	[112,147]
	- Lipid metabolism favors ferroptosis - Oleic acid protects from ferroptosis	[100,113]
	Resistance to	
Downregulation or lack of tumor antigen expression	ICI, adoptive T-cell therapy, RNA vaccine	[208,213,214,215,216]
Alterations in antigen processing pathways	ICI	[217,218,219,220,221]
Loss of HLA expression	ICI, adoptive T-cell therapy Therapeutic vaccine composed of autologous tumor cells and BCG in melanoma	[217,218,219,220,221,222,223,224,225]
	BCG vaccine in bladder cancer	[226]
	Autologous virus-specific T-cell transfer in Merkel cell carcinoma	[227]
Expression of multiple immune checkpoints	Neoantigen vaccine combined with ICI	[228]
TNF-α and IFN-γ antitumoral effects	ICI	[229,230,231,232]
WNT–β-catenin signaling, PTEN loss	ICI, melanoma-peptide/interleukin-12 vaccine	[233,234,235]
“Cold” TME	Therapeutic peptide vaccination	[236]
**Extrinsic paradoxes and resistance**		
Target	Paradoxes	
ECM remodeling	- In the lung, LOX activity is favored by high oxygen concentration - In other organs, LOX activity is induced by hypoxia through the activation of HIF	[148,149,150,151]
CAFs	- Transfer of high-energy metabolites to cancer cells via CAV-1 - Loss of CAV-1 leads to EMT	[123,124,125,126]
	- NF-κB signaling promotes protumor inflammation - Tumor-suppressive function of IKKβ/NF-κB	[122,131]
Neutrophils	- Prevention of pre-metastatic niche formation (via oxidative stress) - Entrapment of circulating cancer cells helping to form the niche (via NETs)	[143,144,145]
	- Toxic to cancer cells in the absence of TGF-β signaling, whereas when the TGF-β pathway is functional, neutrophils are skewed toward an immunosuppressive phenotype	[146]
Tregs	- Activation of neutrophils and macrophages → cytokines → cancer progression - Facilitation of DCs and CTL → antitumor and anti-metastatic effects	[155,156]
TAMs	- Within normoxic tumoral tissue, the M1 response is ensured - In hypoxic regions, an M2 response prevails	[164,165]
TGF-β	- During cancer progression, it acts as a tumor suppressor - In the metastasization process, it induces the EMT and invasion of cancer cells	[59]
Type 1 IFN	- Absence of signaling → enhanced metastatic load and neutrophil accumulation in metastases - Anti-immune program by inducing the expression of immunosuppressive factors (PD-L1, IDO, and IL-10) on DCs → metastasization	[166,167,168,169]
Chemokines	- CXCL2, CXCL8, CXCL4, CXCL9, CXCL10, CXCL12, CCL2, and CXCR2 → proangiogenic - CXCR3, CXCL9, CXCL10, and CXCL4 → anti-angiogenic	[171]
	Resistance to	
Inhibitory receptors (PD1 and/or CTLA-4)	ICI	[237,238]
Production of immunosuppressive cytokines	ICI	[237,239,240]
Macrophages (CSF-1, arginase 1, inducible nitric oxide synthase (iNOS) and ROS)	ICI	[176,241,242,243,244]
Inhibition of systemic and local T-cell activation via Tregs	ICI	[245]
Polarization of local CD4+ T cells, neutrophils, and monocytes	ICI, NY-ESO-1 vaccination	[246,247]
DC function	ICI	[248]
Numbers of Treg cells and MDSCs	ICI, DC-based vaccine, NY-ESO-1 vaccination	[247,249,250]
CAF-mediated regulation of DC proliferation and migration, T-cell infiltration, recruitment of MDCs through ECM stiffness	ICI, cancer vaccines	[54,239,251,252,253]

EMT: epithelial–mesenchymal transition; HIF: hypoxia-inducible factor; ICI: immune checkpoint inhibition; IDO: indoleamine 2, 3-dioxygenase; LOX: lysyl oxidase; PDL: programmed death ligand; TAMs: tumor-associated macrophages; TFs: transcription factors; Tregs: T regulatory cells.

## Data Availability

Not applicable.

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
