# Peer review of "A Vaccine against Cancer: Can There Be a Possible Strategy to Face the Challenge? Possible Targets and Paradoxical Effects"

_vaccines, 2023, doi:10.3390/vaccines11111701_

Round 1

Reviewer 1 Report

The article has several shortcomings and mistakes across its sections. Here's few examples.

Introduction:

The introduction begins with a general overview of cancer vaccines but lacks a clear and concise statement of the article's objectives or hypotheses. It fails to provide a strong rationale for why a vaccine targeting metastasis is necessary, making it challenging for readers to understand the significance of the proposed research. Additionally, the introduction does not mention any specific research questions or hypotheses that the article intends to address, which is crucial for setting clear expectations for the reader.

The Metastatic Process:

The section could be better structured for clarity and flow.

The Tumor Microenvironment:

This subsection lacks a clear transition from the previous section. The discussion on the role of various cells and components in the tumor microenvironment is somewhat fragmented and lacks a cohesive narrative. To improve clarity, the section should provide more context and specific examples of how the tumor microenvironment influences cancer progression.

The ECM:

This subsection introduces concepts like LOX activity and ECM stiffness but fails to provide concrete examples or studies to support these claims. The section also lacks a clear connection to the broader theme of the article, which is the development of a vaccine targeting metastasis. It should better integrate the discussion of the ECM into the overall narrative of the article.

TGF-β:

While this subsection discusses the role of TGF-β in cancer metastasis, it does not clearly connect this discussion to the main theme of the article, which is the development of a metastasis-targeting vaccine.

Epithelial-to-Mesenchymal Transition:

This subsection introduces the concept of epithelial-to-mesenchymal transition (EMT) but does not provide a clear explanation of its relevance to cancer metastasis. It mentions various EMT-related factors but lacks concrete examples or studies that demonstrate their paradoxical effects. The section should better connect the discussion of EMT to the broader theme of the article and provide more context to make it accessible to a general audience.

 CAFs:

This subsection lacks specific references or examples to illustrate the paradoxical effects of CAFs in different contexts. The section could benefit from case studies or research findings that highlight the dual nature of CAFs in promoting and inhibiting cancer progression. Additionally, it should better connect the discussion of CAFs to the main theme of the article.

The Pre-Metastatic Niche:

This section introduces the concept of the pre-metastatic niche but does not provide a clear explanation of its relevance to cancer metastasis or how it relates to the main theme of the article. It mentions immune cells and their role in regulating the pre-metastatic niche but lacks specific examples or studies to support these claims. The section should better integrate the discussion of the pre-metastatic niche into the overall narrative of the article and provide more context to make it accessible to a general audience.

Immunity and Inflammation:

It mentions the polarization of immune cells but does not provide clear connections to the main theme of the article, which is the development of a vaccine targeting metastasis. The section should better connect these discussions to the overarching narrative.

Acquired Resistance to Therapy and TME:

This section briefly mentions the influence of the tumor microenvironment on cancer response to therapies but lacks specific examples or studies to support these claims. It does not clearly connect this discussion to the main theme of the article, which is the development of a metastasis-targeting vaccine. The section should provide more context and specific examples to enhance its relevance to the article's objectives.

Discussion:

The discussion section starts with a general statement about the complexity of cancer but lacks a clear synthesis of the article's key findings or insights. It mentions the need for a new approach but does not provide specific recommendations or strategies for developing a metastasis-targeting vaccine.

Author Response

Q1

Introduction:

The introduction begins with a general overview of cancer vaccines but lacks a clear and concise statement of

the article's objectives or hypotheses. It fails to provide a strong rationale for why a vaccine targeting metastasis is necessary, making it challenging for readers to understand the significance of the proposed research. Additionally, the introduction does not mention any specific research questions or hypotheses that the article intends to address, which is crucial for setting clear expectations for the reader.

A1

Thank you for your useful advice, since it allowed to meliorate very much our paper. In fact, in keeping with our obervations, we modified the Introduction.

Q2

The Metastatic Process:

The section could be better structured for clarity and flow.

A2

Thank you for your useful advice, we changed also the title of this paragraph beyond its content, we would trust that it seems more clear now.

Q3

The Tumor Microenvironment:

This subsection lacks a clear transition from the previous section. The discussion on the role of various cells and components in the tumor microenvironment is somewhat fragmented and lacks a cohesive narrative. To improve clarity, the section should provide more context and specific examples of how the tumor microenvironment influences cancer progression.

A3

Thank you for your useful advice, we linked this paragraph to the previous section and tried to limit its fragmentation making it more narrative.

Q4

The ECM:

This subsection introduces concepts like LOX activity and ECM stiffness but fails to provide concrete examples or studies to support these claims. The section also lacks a clear connection to the broader theme of the article, which is the development of a vaccine targeting metastasis. It should better integrate the discussion of the ECM into the overall narrative of the article.

A4

Thank you for your useful advice, we deepened LOX and ECM stiffness and connected better this section to the theme of article. Moreover, we tried to integrate this section into the overall theme of article.

Q5

TGF-β:

While this subsection discusses the role of TGF-β in cancer metastasis, it does not clearly connect this discussion to the main theme of the article, which is the development of a metastasis- targeting vaccine.

A5

Thank you for your useful advice, we added some sentences and modified others, therefore verifying the conditions in which TGF-β cannot be proposed as possible candidate to design a vaccine that prevents metastasis.

Q6

Epithelial-to-Mesenchymal Transition:

This subsection introduces the concept of epithelial-to- mesenchymal transition (EMT) but does not provide a clear explanation of its relevance to cancer metastasis. It mentions various EMT-related factors but lacks concrete examples or studies that demonstrate their paradoxical effects. The section should

better connect the discussion of EMT to the broader theme of the article and provide more context to make it accessible to a general audience.

A6

Thank you for your useful advice, we modified this section according to your suggestions. We hope that you agree with its overall improvement.

Q7

CAFs:

This subsection lacks specific references or examples to illustrate the paradoxical effects of CAFs in different contexts. The section could benefit from case studies or research findings that highlight the dual nature of CAFs in promoting and inhibiting cancer progression. Additionally, it should better connect the discussion of CAFs to the main theme of the article.

A7

Thank you for your useful advice, we followed you suggestions and the results should convey the notio of dual nature of CAFs.

Q8

The Pre-Metastatic Niche:

This section introduces the concept of the pre-metastatic niche but does not provide a clear explanation of its relevance to cancer metastasis or how it relates to the main theme of the article. It mentions immune cells and their role in regulating the pre- metastatic niche but lacks specific examples or studies to support these claims. The section should better integrate the discussion of the pre-metastatic niche into the overall narrative of the article and provide more context to make it accessible to a general audience.

A8

Thank you for your useful advice, we reitaned useful to report data from literature about this specific topic and we tried to integrate it into overall narrative clarifying our aim that could justify its mention.

Q9

Immunity and Inflammation:

It mentions the polarization of immune cells but does not provide clear connections to the main theme of the article, which is the development of a vaccine targeting metastasis. The section should better connect these discussions to the overarching narrative.

A9

Thank you for your advice, we could not avoid to address this topic because of its critical roles during carcinogenesis. From its evaluation is derived the proposal to stimulate Th1 response as shown at the end of our new version of article. We tried to connect it to overarching narrative.

Q10

Acquired Resistance to Therapy and TME:

This section briefly mentions the influence of the tumor microenvironment on cancer response to therapies but lacks specific examples or studies to support these claims. It does not clearly connect this discussion to the main theme of the article, which is the development of a metastasis-targeting vaccine. The section should provide more context and specific examples to enhance its relevance to the article's objectives.

A10

Thank you for your insightful advices, they allowed to ameliorate the paper and find ectonucleotidases as possible candidates to project a vaccine. So the Section integrate much better into the article theme.

Q11

Discussion:

The discussion section starts with a general statement about the complexity of cancer but lacks a clear synthesis of the article's key findings or insights. It mentions the need for a new approach but does not provide specific recommendations or strategies for developing a metastasis-targeting vaccine.

A11

Thank you for your useful advice too, your suggestions to modify the discussion allowed article to achieve its goal in specifying possible candidate to project a cancer vaccine; the previous version, focusing above all on paradoxical effects/behaviors, would not have provided an useful help as you said.

Thank you for your valuable work.

Reviewer 2 Report

A Vaccine against Cancer: Can There Be a Possible Strategy to Face the Challenge?

The review titled “A Vaccine against Cancer: Can There Be a Possible Strategy to Face the Challenge?

According to authors, the proposal of this review is to discuss about a vaccine that administered to patients can preventing metastasis. Metastasis represents the event that induce the death, thus preventing it could mean transforming cancer in a chronic disease. The authors discuss about the presence of the paradoxical actions of different mechanisms, pathways, molecules and immune and non-immune cells characteristic of tumor microenvironment at the primary site and pre-metastatic niche. They conclude that changes in the current cancer vaccine concept are needed, with the increasing hope to eradicate cancers, the first cause of death in recent times.

I have found this review suitable and with high potential to be cited in the field of immunotherapy of cancer, however my recommendation is to publish after minor revisions.

4. Immunity and inflammation

I consider this could be resumed in a figure due the importance of chronic inflammation and some persistent infections and its relation to leading a transformation.

Figure 1 Figure 1. Paradoxical concepts about cells and mediators within the TME, is very fully in the present form, but is difficult to understand. I could suggest to modify or to separate in two figures in order to shown readers figures more easy to follow?

Author Response

According to authors, the proposal of this review is to discuss about a vaccine that administered to patients can preventing metastasis. Metastasis represents the event that induce the death, thus preventing it could mean transforming cancer in a chronic disease. The authors discuss about the presence of the paradoxical actions of different mechanisms, pathways, molecules and immune and non-immune cells characteristic of tumor microenvironment at the primary site and pre-metastatic niche. They conclude that changes in the current cancer vaccine concept are needed, with the increasing hope to eradicate cancers, the first cause of death in recent times.

I have found this review suitable and with high potential to be cited in the field of immunotherapy of cancer, however my recommendation is to publish after minor revisions.

We thank your for the overall appreciation of our work.

4. Immunity and inflammation

I consider this could be resumed in a figure due the importance of chronic inflammation and some persistent infections and its relation to leading a transformation.

We agree with the Reviewer that a figure on the role of inflammation/immunity in this context would be useful, thus we provide a novel Figure 3 on this topic. Although persistent infections are of course of interest in the chronic inflammation associated with cancer progression, however this topic would be far from our specific work and are not represented in the figure.

Figure 1 Figure 1. Paradoxical concepts about cells and mediators within the TME, is very fully in the present form, but is difficult to understand. I could suggest to modify or to separate in two figures in order to shown readers figures more easy to follow?

We thank the Reviewer for this consideration, thus we split the Figure 1 in two novel figures. Furthermore we have also introduced a color code to make more easily understandable pro- and anti-tumor effects.

Reviewer 3 Report

1) The topic of the Review manuscript submitted is of interest to scientific community. However the manuscript needs extensive rewriting and change in format. English language needs extensive reviewing and modification.

2) Title of the manuscript needs to be changed and more focused to the content of the manuscript. Current manuscript title is vague and do not point to the specific topic discussed in the manuscript.

3) The concept discussed in the manuscript is already know to scientific community however any specific solution is not there . The authors needs to focus on specific cancer type (Breast, Lung etc.) and not the pathways related to cancer prevention in different type of cancers and try to implement them for all type of cancers. This will help in understanding the cancer therapy options in Primary verses Metastatic cancer in a specific cancer type and will be more informative and easy to follow.

4) Tables indicating what are the options available and what works in a Metastatic disease state for a cancer type may be a valuable addition to the manuscript. 

5) The manuscript has been written very broadly with the the use of terms that are  unscientific such as "no one thought or looked at the problem etc etc", the real challenge is how to implement a therapy for cancer that can be also implemented for metastatic cancer. The manuscript needs to focus at the problem directly without any personal or groups opinion/statement.

English needs to be extensively improved.

Author Response

1) The topic of the Review manuscript submitted is of interest to scientific community. However the manuscript needs extensive rewriting and change in format. English language needs extensive reviewing and modification.

We thank the Reviewer for her/his appreciation of our work. English language has been extensively reviewed.

2) Title of the manuscript needs to be changed and more focused to the content of the manuscript. Current manuscript title is vague and do not point to the specific topic discussed in the manuscript.

We thank the Reviewer for her/his comment on the manuscript title, which now reads as “A Vaccine against Cancer: Can There Be a Possible Strategy to Face the Challenge? Possible Targets and Paradoxical Effects.” We believe that now it is more focused on the content of the manuscript.

3) The concept discussed in the manuscript is already know to scientific community however any specific solution is not there . The authors needs to focus on specific cancer type (Breast, Lung etc.) and not the pathways related to cancer prevention in different type of cancers and try to implement them for all type of cancers. This will help in understanding the cancer therapy options in Primary verses Metastatic cancer in a specific cancer type and will be more informative and easy to follow.

We thank the Reviewer for her/his comment on the manuscript content and organization. The aim of this article was to review the overall involvement of cells, pathways and molecules in primary growth and metastatic sites to give an appreciation of those possible targets and paradoxical effects that ultimately might give rise to new cancer vaccines. Although not specifically focussed on one specific cancer type, many detailed examples of cancer arising in different organs are presented.

4) Tables indicating what are the options available and what works in a Metastatic disease state for a cancer type may be a valuable addition to the manuscript.

We thank the Reviewer for her/his valuable comment, now we present in the new version of the Discussion Section the vaccine targets we believe that should be further investigated in each cancer type, thus another table could be redundant.

5) The manuscript has been written very broadly with the the use of terms that are unscientific such as "no one thought or looked at the problem etc etc", the real challenge is how to implement a therapy for cancer that can be also implemented for metastatic cancer. The manuscript needs to focus at the problem directly without any personal or groups opinion/statement.

We thank the Reviewer for her/his useful comment. We have deleted those sentences that are not conveying an objective vision of the topic presented in the article.

Comments on the Quality of English Language

English needs to be extensively improved.

English language was extensively reviewed.

Round 2

Reviewer 1 Report

The Manuscript has been improved.